# Ferroptosis and Radiotherapy in Lung Cancer

**DOI:** 10.3390/cells14231927

**Published:** 2025-12-04

**Authors:** Prem C. Patel, Eva M. Galvan

**Affiliations:** Department of Radiation Oncology, University of Texas at San Antonio, San Antonio, TX 78229, USA; patelp22@livemail.uthscsa.edu

**Keywords:** ferroptosis, radiotherapy, radiation therapy, lung cancer

## Abstract

Background: Lung cancer (LC) is a leading cause of cancer mortality worldwide. While radiotherapy (RT) has been a lasting cornerstone of LC management, there are concerns due to tumor radioresistance and unintended damage to surrounding healthy tissue. Ferroptosis is a recently described mechanism of programmed cell death which has potential to serve as a complementary adjunct to facilitate RT-based LC treatment. Objectives: This review is a comprehensive overview of ferroptosis in the broader context of synergism with RT for LC. Summary: Ferroptosis is essentially driven by intracellular iron overload, which drives the formation of reactive oxygen species, ultimately resulting in membrane instability and cell death. LC lines have been shown to exhibit a heterogeneous mix of pro- and anti-ferroptotic changes. RT shows promise as a potential ferroptosis inducer, especially when complemented with pharmacologic agents such as erastin. Conclusions: Ferroptosis represents a promising modern adjunct to a traditional therapeutic strategy. Future work should focus on rigorous dosage standards to avoid unintended toxicity, repurposing of currently available drugs into ferroptosis inducers, and establishment of safety protocols to begin the pathway towards clinical studies.

## 1. Introduction

Lung cancer (LC) is a leading cause of cancer and cancer mortality, accounting for about 219,000 new cases in the United States in 2023 and 132,000 deaths in the United States in 2023 [1]. Radiotherapy (RT) has been a cornerstone management tool for decades, providing treatment for about 40% of LC patients [2]. RT offers a minimally invasive, convenient, and efficacious tool for LC management, although optimal benefit is hindered primarily by two barriers: acquired tumor cell resistance and unintended damage to surrounding healthy tissue [3,4,5].

Ferroptosis is a recently described modality of regulated cell death characterized by iron-dependent accumulation of lipid peroxides and reactive oxygen species (ROS) [6]. The process is tightly regulated by several molecular pathways, and has distinct morphologic, genetic, and biochemical features which separate it from other methods of cell death like apoptosis and necrosis. Disruption of these systems by genetic, pharmacologic, or environmental stressors can precede ferroptosis [7].

RT functions as a stressor by inducing the accumulation of ROS within tumor cells. In addition to directly causing DNA damage, RT-derived ROS also contribute to the peroxidation of lipids, triggering ferroptosis in targeted cancer cells [8]. This effect can be amplified through combination of RT with ferroptosis inducer treatment, providing a synergistic mechanism for enhanced cancer cell death, an effect which is particularly valuable in overcoming radioresistant tissues [9]. Despite the promise of RT-induced ferroptosis, many lung tumors can adapt by upregulating resistant pathways [10].

Although ferroptosis can be a powerful tool for enhancing cytotoxicity in lung cancer treatment, it is not exclusively beneficial. Normal lung tissue is also susceptible to RT-induced ferroptosis, resulting in unwanted damage to healthy cells [11]. Likewise, systemic therapy with ferroptosis inducers can increase normal cell death in areas not receiving radiation damage [12]. This highlights the critical need for balance in treatment design to allow for optimal therapeutic ratio to minimize potential toxicity.

This review provides a comprehensive overview of ferroptosis biology within the context of lung cancer, highlights the synergism between ferroptosis and RT, and summarizes opportunities and challenges for clinical treatment. We aim to showcase the potential for ferroptosis modulation as a viable avenue for improving RT efficacy in LC treatment.

## 2. Mechanisms of Ferroptosis

### 2.1. Iron and Lipid Metabolism

Iron plays a critical role in DNA synthesis and repair, oxygen transport, cellular respiration, and other critical metabolic and enzymatic functions, although it can become toxic when poorly regulated. Iron circulates through the bloodstream as ferric iron (Fe^3+^) attached to the protein transferrin. It enters cells primarily via the transferrin receptor 1 (TfR1). Once in the cell, iron is reduced to its ferrous form (Fe^2+^) and stored as part of the cytosolic labile iron pool (LIP). While the LIP is critical for the cellular processes discussed above, iron abundance drives the Fenton reaction (Fe^2+^ + H_2_O_2_ → Fe^3+^ + •OH + OH^−^). The hydroxyl radical (•OH) generated from this reaction is highly reactive, and initiates lipid peroxidation [13]. In addition to uptake of ferric iron from the bloodstream, other contributors to the LIP include autophagic destruction of ferritin (ferritinophagy), heme degradation by heme oxygenase 1 (HO-1), and release of free iron from the mitochondria [14,15,16].

Ferroptosis is specifically linked to the peroxidation of polyunsaturated fatty acids (PUFAs), which are especially vulnerable to oxidative attack due to the presence of bis-allylic carbons, which have low activation energy for hydrogen loss by ROS [17]. Free cytosolic PUFAs are first activated into acyl-CoA derivatives by Acyl-CoA synthetase long-chain family member 4 (ACSL4), and are then incorporated into membrane phospholipids via lysophosphatidylcholine acyltransferase (LPCAT3) [13,18]. Due to the high density of these compounds within the membrane, a lipid radical generated from the initial reaction with a hydroxyl radical can then abstract another hydrogen atom from an adjacent PUFA, propagating this process into a chain reaction which is difficult to control without sufficient antioxidant mechanisms. Lipid peroxides then accumulate, destabilizing plasma membrane integrity resulting in cell death [19], as detailed in Figure 1.

### 2.2. Antiferropoptic Mechanisms

Cells possess multiple mechanisms to counter lipid peroxidation, the most relevant one in ferroptosis being the glutathione (GSH)–glutathione peroxidase 4 (GPX4) axis. GPX4 is a selenoprotein that reduces lipid hydroperoxides (L-OOH), like those generated from PUFA chain reactions, and reduces them to stable lipid alcohols (L-OH), preserving membrane integrity and protecting from ferroptosis-induced cell death. GPX4 requires a reducing agent cofactor in the form of GSH [20,21]. In addition to this axis, there are several other mechanisms which can provide “backup” production to lipid peroxidation. The ferroptosis suppressor protein 1 (FSP1) regenerates reduced coenzyme Q10 (CoQ10H_2_, ubiquinol) which acts as a direct lipid radical scavenger. Similarly, the mitochondrial protein dihydroorotate dehydrogenase (DHODH) and the cytosolic protein GTP cyclohydrolase-1 (GCH1) support resistance by producing reduced CoQ10 and tetrahydrobiopterin (BH_4_), another protective reducing agent [22]. These complementary pathways form a multilayered protection system against lipid peroxidation and ferroptosis.

### 2.3. Biochemical and Morphological Features of Ferroptosis

Ferroptosis is a mechanism of cell death unique to classical modalities such as apoptosis or necrosis. For instance, ferroptosis is characterized by impaired membrane integrity and lacks typical apoptotic markers such as nuclear condensation, DNA condensation, and cell shrinkage. Ferroptotic cells also display unique morphological features such as shrunken mitochondria with reduced or absent cristae and ruptured outer membranes [23]. In contrast to necrosis which involves swelling and membrane rupture secondary to ion imbalance, ferroptosis is a regulated process specifically initiated by iron-dependent lipid peroxidation [24]. Previously discussed biochemical markers such as elevated lipid ROS and depletion of GSH are also unique indicators. Ferroptosis is a relatively new concept, first coined in 2012 and there are a wide variety of ongoing research efforts to gain insights into mechanistic details, biochemical hallmarks, and clinical implications [25].

### 2.4. Ferroptosis-Related Changes in Lung Cancer

Since ferroptosis represents a novel mechanism for regulated cell death that contrasts traditional apoptotic pathways, it has strong relevance in cancer biology, both as an effect of and potential complement to traditional treatment strategies. Many neoplasms develop resistance to apoptosis through inactivating mutations in tumor suppressor genes, overexpression of anti-apoptotic proteins such as BCL-2, or alterations in cellular caspase signaling [26,27]. Ferroptosis provides an alternative route for eliminating these resistant cells. Tumor cells often have dysregulated iron metabolism, which may predispose them to ferroptosis. Elevated expression of TfR1 and down regulation of ferroportin, an iron exporter, lead to intracellular accumulation of the LIP, enhancing ROS development through Fenton chemistry [28].

There have been many ferroptosis-related changes within LC. One study reported that lower expression of ACSL4 was associated with decreased survival, increased tumor size, and higher metastatic rates of malignant pulmonary nodules. This suggests that ACSL4 downregulation may aid tumor cells in avoiding ferroptosis, since lower expression of ACSL4 depletes the cell membrane of reactive PUFANs and decreases susceptibility to peroxidative damage [29]. Conversely, the transcription factor NFE2L2 (NRF2), activated in many non-small-cell lung cancers (NSCLCs) contributes to ferroptotic resistance by upregulating ferritin, an iron sequestering protein which depletes the LIP, and other antioxidant genes including GPX4 and SLC7A11, a cystine/glutamate membrane transporter required for GSH production [30]. Scott et al. (2022) found that many cancer types, including lung, have exhibited metabolic adaptations by incorporating more monounsaturated fatty acids (MUFAs) in their membranes rather than unstable PUFAs to protect from lipid peroxidation, effectively reducing oxidative damage and ferroptotic susceptibility [31]. It is also important to understand that not all LCs are the same: Bebber et al. (2021) found that within small-cell lung cancer (SCLC), non-neuroendocrine tumors are particularly sensitive to ferroptosis, while neuroendocrine tumors display more resistant due to increased reliance on the TRX antioxidant pathway [32].

Overall, lung cancers exist at an intersection between pro-ferroptotic and anti-apoptotic mechanisms. A dynamic mix of iron metabolism, lipid composition, and tumor genotype all influence not only vulnerability to ferroptosis, but also the ability to exploit it through oncological treatment. This opens opportunities for ferroptosis-targeted therapies, especially when applied in conjunction with traditional approaches like RT.

## 3. Implications of Ferroptosis in Radiotherapy

### 3.1. Radiotherapy as a Ferroptosis Inducer

Over the years, RT has remained a staple of LC, with 40% of all patients receiving it during their treatment course [2]. To ensure tumor death and sparing of healthy cells, ionizing radiation beams are precisely coordinated in conjunction with the patient’s imaging studies to target only the tumor [33]. These beams are delivered from multiple angles, allowing tumor cells to receive a high dose while minimizing exposure to healthy tissue. Although its classic mechanism is radiation-induced DNA double-strand breaks and generation of ROS which destroy macromolecules critical for cell survival, there is emerging evidence that suggests RT can also trigger ferroptosis [34]. This can potentially be harnessed to overcome radioresistance but conversely may contribute to treatment toxicity [35].

Recent mechanistic studies have identified several pathways in which RT primes tumor cells for death by ferroptosis. Firstly, common ROS generated from RT such as hydroxyl radical and superoxide (O_2_^−^), in addition to damaging DNA, directly drive the peroxidation of membrane phospholipids, tipping the balance towards ferroptosis [9,36]. Next, RT-induced DNA damage activates the repair protein ATM kinase, which studies show downregulates SLC7A11, reducing cystine import and thus intracellular GSH levels. RT has also been shown to directly decrease GPX4 levels [35]. These mechanisms compromise the GSH-GPX4 axis, the body’s primary defense against ferroptosis. Radiation exposure also increases ACSL4 activity, enriching the membrane with PUFAs [37].

Ferroptotic susceptibility may also be influenced by RT dosing. Lang et al. (2019) found that a single fraction of 10 Gray (Gy) radiation induced greater lipid peroxidation compared to 5 fractions of 3 Gy in B16F10 melanoma cells [37]. Another preclinical study of nasopharyngeal carcinoma found that ferroptotic markers (ACSL4, PTGS2, and IREB2) were significantly higher compared to baseline at 10 Gy, whereas many were not significantly increased at 2 Gy [38]. Despite these novel investigations, there seems to be a relative lack of studies on dose-dependence of ferroptosis in LC.

### 3.2. Ferroptosis Modulators as Radiosensitizers

Despite the many practical benefits and effectiveness of RT in LC, there is a growing concern about radioresistance amongst tumor cells. While RT alone can induce cell death via ferroptosis through mechanisms described above, some persistent LC lines upregulate antioxidant defenses like NFR2, SLC7A11, GPX4, and FSP1, protecting these cells from death and decreasing treatment efficacy [39,40]. Because RT may not reliably induce ferroptosis in these resistant tumors, there have been many preclinical studies testing the use of ferroptosis inducers used in combination radiation, with promising results in non-small-cell lung cancer (NSCLC) lines, as listed in Table 1.

Perhaps the most commonly discussed ferroptosis inducer is the small quinazoline erastin, a blocker of cystine uptake which depletes intracellular GSH-GPX4 activity. Shibata et al. (2019) found that erastin enhanced the cytotoxic effects of X-ray irradiation in the human NCI-H1975 line of pulmonary adenocarcinoma transplanted into mice [41]. In these xenografts, combined treatment suppressed tumor growth more than either treatment alone. Sorafenib is a chemotherapeutic kinase inhibitor that has been shown to also act as a ferroptosis inducer. In fact, Li et al. (2020) found that sorafenib and erastin both triggered ferroptosis of cisplatin-resistant NSCLC cells in a mouse model, and effectively inhibited cell growth in vivo [42].

RSL3 is a direct GPX4 inhibitor which bypasses the need to deplete cystine or GSH stores. In vitro, RSL3 sensitized LC cells to RT, showing higher lipid ROS levels and decreased clonogenic survival [9]. Tubastatin A, an inhibitor of histone deacetylase 6 (HDAC6), is another GPX4 destabilizer, which has been found to assist in overcoming established radioresistance in murine lung tumor models [43].

Other molecules which have shown potential for the induction of ferroptosis are NRF2 pathway inhibitors (e.g., ML385) and the telomerase inhibitor BIBR1532 [44,45]. Even common agents like acetaminophen have shown to synergize with erastin in xenograft models [46]. Overall, there are many new ferroptosis inducers, and more studies should be performed to evaluate their efficacy in negating radioresistance in LC.

Additional inducers have also demonstrated radiosensitizing potential in preclinical cancer models. Fin56 and FINO_2_ are novel molecules which induce ferroptosis by prompting GPX4 degradation and iron oxidation, respectively [47,48,49]. Dihydroorotate Dehydrogenase (DHODH) inhibitors, such as brequinar, have also expressed potential as ferroptosis inducers by suppressing mitochondria antioxidant activity, allowing the progression of ROS accumulation and intensifying ferroptotic stress [50]. Likewise, artesunate, an antimalarial, induced ferroptosis in a diffuse large B-cell lymphoma model by inhibiting the antioxidant enzymes PRDX1 and PRDX2 [51]. Lastly, HMG-CoA reductase inhibitors (statins), commonly prescribed for hyperlipidemia, have been shown to induce ferroptosis through multiple mechanisms including indirect inhibition of the PD-L1 pathway via action on interleukin-enhancing binding factor 3 (ILF3) and downregulation of antioxidant defenses such as GSH, GPX4, and SLC7A11 [52,53].

### 3.3. Pathway Interactions and Genetic Content

Every cancer cell line is different. Two individuals with the same LC subtype may have vastly different therapeutic responses depending on minute variations in individual gene mutations, epigenetic modifications, and protein activity. To that end, the extent to which RT induces ferroptosis depends greatly on the specific tumor’s molecular background. For example, the wild-type tumor suppressor protein p53 can repress SLC7A11, sensitizing to ferroptosis, while p53-p21 interactions may protect from ferroptosis in some circumstances [54,55]. Therefore, tumors with mutations in the TP53 gene may exhibit an altered ferroptotic phenotype, potentially increasing susceptibility to RT.

The KEAP1 protein is a repressor of NRF2 that is frequently inactivated in NSCLC. This mutation leads to protection from ferroptotic mechanisms secondary to NFR2-upregulated antioxidant defenses such as ferritin (FTH1, FTL), SCL7A11, and HO-1 [56]. In KEAP1-deficient LC models, FSP1 is upregulated and confers ferroptotic resistance [57]. Importantly, inhibiting FSP1 in these mutated lines restores lipid peroxidation and radiosensitivity.

The oncogenic KRAS GTPase, commonly mutated in pulmonary adenocarcinoma, has also been implicated in intracellular redox changes, though specific data are mixed [58]. KRAS-mutated cells often exhibit elevated FSP1 expression, reduced lipid peroxidation, and increased ferroptosis resistance, although some KRAS-driven lines may promote increased iron uptake and sensitization to ferroptosis [59,60]. This highlights how specific redox signatures and ferroptotic responses may vary based on specific molecular changes.

Overall, the literature suggests that LC is highly variable, and patient stratification based on genetic markers or metabolic markers is crucial when designing and implementing ferroptosis-targeted radiosensitization strategies.

### 3.4. Radiation-Induced Ferroptosis and the Tumor Microenvironment

A tumor microenvironment (TME) is a complex and dynamic network of non-cancerous cells, extracellular matrix components, signaling molecules, and blood vessels that surround and interact with tumor cells [61]. The components of the TME constantly interact with the tumor and contribute significantly to its behavior, actively influencing growth, metastasis, and response to therapy. Beyond causing intrinsic tumor cell death, ferroptosis may have important role in shaping the TME.

Once cancer cells die, they release intracellular molecules such as high-mobility group box 1 (HMGB1), adenosine triphosphate (ATP), and oxidized phospholipids, which can act as danger-associated molecular patterns (DAMPs) [62]. These molecules stimulate dendritic cell maturation, macrophage phagocytosis, and CD8+ T cell proliferation. This adaptive, anti-tumor immune response can boost the effectiveness of immunotherapies. Similarly, RT-induced double stranded DNA damage can generate cytosolic DNA fragments that trigger the cGAS-STING DNA-sensing pathway, leading to type I interferon release and further immune activation [63]. Importantly, recent evidence suggests that ferroptotic cell death can amplify this signaling. Examples include post-ferroptotic release of peroxidized membrane lipids, suppression of regulatory T cells, depletion of M2-Type Tumor-Associated Macrophages (TAMs), and conversion of inactive or “cold” TMEs [64,65].

These processes are examples of how RT-induced ferroptosis not only contributes directly to tumor cell killing but also shapes the surrounding environment in ways that may boost systemic anti-tumor activity. Although the precise mechanisms and clinical implications of these interactions remain under active investigation, it is increasingly evident that ferroptosis lies at a crucial intersection between intracellular cytotoxicity and microenvironment remodeling. Recognizing this dual role provides a strong foundation for exploring combined strategies that leverage RT and ferroptosis induction in the treatment of LC.

## 4. Clinical Integration and Challenges

Although ferroptosis has been well-established in multiple preclinical models, its clinical application in LC remains greatly limited. There has been a relative lack of reporting of ferroptosis inducers in conjunction with RT for human LC treatment. One clinical trial (NCT00543335) that began in 2008 did intend to report the efficacy of sorafenib in NSCLC, although it was terminated in 2018 [66]. Furthermore, the goal of this trial was likely unrelated to sorafenib’s potential as a ferroptotic inducer, as this is not mentioned in the trial webpage and this trial began before the term ferroptosis was coined. We have been unable to find a publication or clinical trial that specifically seeks to investigate the relation of ferroptosis and RT in a human model, highlighting a critical gap. Despite lack of current data, there are several features of LC management strategies in which ferroptosis-based radiosensitization could potentially become well-integrated.

### 4.1. Patient Stratification and Biomarkers

As previously discussed, there are many variant genetic and molecular markers which may reflect tumor sensitivity to ferroptotic induction. Therefore, clinicians should carefully evaluate these indicators before beginning treatment. Thankfully, many of these measures are already examined in traditional clinical practice. For example, mutations in TP53, the KEAP1/NFR2, and KRAS (all of which have been discussed above) are already reported on many commercial next-generation sequencing (NGS) panels used in oncology [67,68]. This provides an easy pathway for some ferroptosis-targeted strategies to be integrated into existing infrastructure without the need for extensive additional testing. However, there is a lack of validated assays for many other ferroptosis-specific factors, like ACSL4 [69].

In addition to biomarkers used to evaluate a patient’s suitability for ferroptotic treatment, large-scale implementation will likely also require real-time biomarkers to evaluate treatment response. Some promising markers are malondialdehyde (MDA) and 4-hydroxynoneal (4-HNE) [70]. These products of lipid peroxidation are detectable in patient plasma, and have been explored as indicators of oxidative stress events such as ferroptosis. Lipid ROS imaging probes are also under early investigation [71]. Incorporating these tools could allow clinicians to monitor tumor response to ferroptotic inducers alongside RT and adjust treatment dosing or frequency as needed.

### 4.2. Balancing Efficacy and Toxicity

While ferroptosis may hold strong potential to potentiate the anti-tumor activity of RT, as with many other cancer therapies, it may also underlie toxicity to healthy lung tissue. One of the most severe complications of lung-focused RT is radiation-induced lung injury (RILI), which can manifest pneumonitis or fibrosis [72]. RILI affects between 5–20% of patients depending on dose and modality. Multiple preclinical studies have demonstrated that ferroptosis of lung tissue directly contributes to RILI [73,74,75]. This raises concerns that combining ferroptotic and RT treatment could exacerbate this toxicity.

Outside of the lungs, systemic administration of ferroptosis inducers has raised important concerns regarding multi-organ toxicity. Ferroptotic stress has been implicated in mouse renal tubular injury by Friedmann Angeli et al. (2016) and hepatocyte damage by Lőrincz et al. (2015) and Yamada et al. (2020) [76,77,78]. Such effects may be exacerbated if ferroptosis inducers are used in conjunction with other systematic anticancer treatments like chemotherapy. This underscores a critical need for delivery methods that maximize tumor-specific damage while sparing normal tissue.

One potential strategy is targeted drug delivery. Nanoparticle carriers, liposomes, and antibody–drug conjugates (ADCs) are tumor-selective delivery methods which have been used to carry ferroptosis inducers [79,80]. Nguyen et al. (2022) developed an ADC containing the GPX4 inhibitor RSL3, providing a strong proof-of-concept for this novel mechanism [81]. Similarly, Liu et al. (2023) embedded liposomes with PEGylated iron oxide nanoparticles, which showed efficient ferroptosis induction in murine breast cancer models [82].

Additionally, physicians could maximize spatial and temporal control by administering ferroptotic inducers during RT treatment. This approach may induce the preferential accumulation in tumor regions already undergoing oxidative stress from RT while minimizing exposure to surrounding untreated tissue. Bae et al. (2024) reported a synergistic benefit in combined ferroptotic inducing nanoparticles and X-ray irradiation therapy [83].

### 4.3. Drug Accessibility

Several ferroptosis inducers are already available for other uses. For example, sulfasalazine, a SLC7A11 inhibitor, is approved for rheumatoid arthritis and has shown to enhance RT efficacy in colorectal and glioma models [84,85]. Similarly, sorafenib has been approved for treatment liver and renal cancers, and while this drug is not currently used specifically for is ferroptotic mechanism, this existing governmental approval highlights a promising pathway forward for drug repurposing [86]. Demonstrated safety profiles and ability to induce cell death in preclinical models suggest strong potential for repurposing into ferroptotic-based strategies.

### 4.4. Integration with Other Treatment Modalities

Although RT is a proven tool for reducing LC mortality and metastasis, it is rarely administered in isolation. Many patients concurrently receive systemic treatment such as platinum-based chemotherapy drugs or immune checkpoint inhibitors [87,88]. For ferroptosis inducers to be truly integrated within the greater realm of LC care, their integration into these established regiments must be considered.

Traditional cytotoxic drugs like the platinum analogues cisplatin and carboplatin are a mainstay of NSCLC treatment. These compounds, while primarily exerting cytotoxicity through DNA crosslinking, also generated oxidative stress and deplete GSH reserves [89]. There is potential synergy between ferroptosis inducers like erastin and RSL4 with these platinum analogs to overcoming resistance in refractory lines. For instance, Liang et al. (2021) found that cisplatin synergizes with the erastin analog PRLX93936 to enhance cell death in NSCLC lines [90]. Another standard therapy for advanced NSCLC is immune checkpoint blockade with programmed cell death protein 1 (PD-1) and programmed cell death ligand (PD-L1) inhibitors [91,92]. These treatments amplify endogenous T-cell activity against tumor cells, and ferroptosis may indirectly enhance this effect. Combining ferroptosis inducers with these checkpoint inhibitors has shown additive tumor control in melanoma and colorectal models although this has not yet been investigated in LC. These data suggest that ferroptosis could be used as a multiplier for systemic therapy in addition to targeted RT by exploiting oxidative stress and increasing tumor immunogenicity [93,94].

## 5. Knowledge Gaps and Future Directions

### 5.1. Limitations of Current Models

Most current studies on the feasibility of ferroptosis inducers used alongside RT examine in vitro cell lines and xenograft models. While these do provide valuable insights in treatment feasibility, they do not replicate the vast heterogeneity of human LC nor the complexity of the human lung microenvironment. A critical next step is the formation of comprehensive in vivo models that more accurately reflect LC burden in humans. Future study designs should include elements such as orthotopic tumor localization, intact animal immune systems, and clinically relevant RT dosing schedules [95]. These additions would allow for smoother transitional efforts, especially when examining interactions between ferroptosis, RT, and the TME.

As discussed earlier, preclinical models show that ferroptosis of lung cells contributes to RILI, showing how lung-targeted ferroptotic treatment represents a double-edged sword which must be optimized to maximize tumor death while avoiding severe complications. There are currently no standardized criteria for an “acceptable” level of ferroptosis in normal lung cells during cancer treatment. Thus, future longitudinal animal studies should be conducted to establish therapeutic windows by comparing conventional and high-dose RT regiments. Defining these boundaries will be essential for guiding dose selection, delivery methods, and patient monitoring strategies.

Despite the demonstration that ferroptosis shapes the TME, it is unknown if these effects are transient or sustained and whether they differ across LC subtypes. Future studies could utilize single-cell sequencing (profiling gene expression at the cellular level) and spatial transcriptomics (analysis of gene expression pattens in their exact physical location within tumor tissue) to map immune cell population after ferroptosis induction [96]. Understanding the longevity and subtype-specificity of these immune effects is critical, as sustained modulation of the TME is likely to translate into meaningful improvement in radio sensitivity and long-term tumor control. Exploration of inter-tumor variability, especially between “cold” versus “hot” TMEs could help guide patient-specific combined treatment strategies.

### 5.2. Safety and Translation to Human Research

Although human research is an exciting prospect for the establishment of ferroptosis modulation in LC, it is essential to establish a safety profile of inducers both as standalone agents and alongside RT. There should be rigorous acute and chronic toxicology studies across multiple organ systems such as hepatic, renal, pulmonary, and neurologic tissues. Such reports will be essential in determining the potential for off-target effects, cumulative toxicity, and long-term effects. Another area of interest is pharmacokinetics. Detailed studies should be performed to understand parameters including absorption and excretion of ferroptosis inducers, as well as tissue penetration and accumulation, particularly in irradiated regions. These insights can inform whether advanced delivery systems are warranted to improve tumor specificity and remove systemic exposure.

Only after a favorable safety profile has been established within multiple animal models can ferroptosis be considered for phase I clinical trials in humans [97]. Early human studies would likely focus on dose-modulation to determine maximum tolerated dose and safety of coadministration with RT and other common cancer therapies. Once these foundational element are established, future studies could focus on exploring optimal dosing regiments, patient selection strategies, combinations with other cancer therapies, and long-term outcomes including treatment response, toxicity, and survival. As previously discussed, nanoparticles offer a promising avenue for translational feasibility. These carriers can enhance tumor specific delivery while minimizing systemic toxicity [98]. Furthermore, these particles can be engineered to meet specific tumor environment characteristics such as acidic pH, high ROS levels, or alternative membrane transporter expression [82,99].

### 5.3. Biomarkers and Precision Integration

Another barrier to clinical translation of ferroptosis-based radiosensitization strategies is the absence of validated biomarkers that can reliably assess treatment response, toxicity risk, and patient suitability. As previously discussed, lipid peroxidation products such as MDA and 4-HNE are strong candidate markers, although their predictive power and reproducibility remain untested. Similarly, while emerging imaging probes for ROS such as the label-free Raman Spectroscopy tool show promise for evaluating treatment response, large-scale validation is needed [100].

Aside from real-time monitoring, biomarker development should extend to baseline patient stratification. While NGS panes routinely detect tumor-specific mutations, they often do not include integrated molecular markers or signatures such as metabolic, lipidomic, and iron-regulatory factors which are also key components of determining ferroptotic vulnerability. Future studies should pilot the addition of these elements to traditional sequencing techniques to generate predictive algorithms for identifying a tumor’s potential sensitivity to ferroptotic and RT treatments, determining which patients are most likely to benefit from this strategy.

## 6. Discussion

RT has long been a major tenet in the management of LC, yet its potential is often limited by the challenges of tumor radioresistance and collateral injury to healthy tissue. Ferroptosis introduces a modern avenue for amplifying this classic modality. By exploiting oxidative stress and lipid peroxidation, RT-induced ferroptosis offers the potential to bypass traditional resistance pathways, sensitize refractory tumors, and reshape the TME in favor of anti-tumor immunity. These qualities highlight the promise of ferroptosis as a useful complement to RT in the evolving landscape of LC treatment.

Preclinical studies support strong synergy between RT and ferroptosis inducers. These models demonstrate that compounds such as erastin, sorafenib, and GPX4 inhibitors can potentiate radiation efficacy, including in resistant tumor cell lines. Concurrently, RT itself primes cells for ferroptotic death through ROS accumulation, depletion of antioxidant defenses, and membrane remodeling. Together, these mechanisms form the basis of a therapeutic capability to overcome traditional barriers to treatment. Beyond direct tumor effects, ferroptotic signaling within the TME may enhance systemic control by stimulating dendritic cell activation, cytotoxic T-cell proliferation, and immunogenic reconstitution of “cold” tumors. These interactions also raise the possibility of combining ferroptosis-associated RT with other cancer therapies such as immune checkpoint inhibitors or chemotherapeutic agents for further tumor control.

Despite the promise of ferroptosis inducers, many questions remain regarding their translation to a human population. Ferroptosis is not tumor-exclusive; healthy lung tissue is also vulnerable, and unregulated activity may predispose patients to RILI or other complications. Systemic delivery of ferroptosis inducers also raises concern about toxicity to other organ systems, emphasizing the need for targeted delivery mechanisms and carefully constructed dosing schedules. Additionally, the heterogenic nature of LC warrants personalization; there can be no “one size fits all” approach to ferroptotic treatment. Genetic alterations in TP53, KEAP/NF2, and KRAS, as well as metabolic and lipidomic signatures, should be screened to help select the most appropriate candidates. Furthermore, the development and implementation of validated biomarkers for both baseline stratification and treatment response monitoring will be critical for effective integration.

The path forward requires robust in vivo models, rigorous toxicology analysis, and early clinical trials to establish safety and efficacy. With the establishment of these foundations, ferroptosis could have the potential to transform RT from a primarily DNA-damaging therapy into a multifaceted approach that harnesses oxidative cell death and immune activation. While there is much to be learned, the convergence of these two therapies represents a new frontier in LC treatment: one that offers hope for overcoming resistance, improving tumor control, and ultimately extending survival.

## Figures and Tables

**Figure 1 cells-14-01927-f001:**
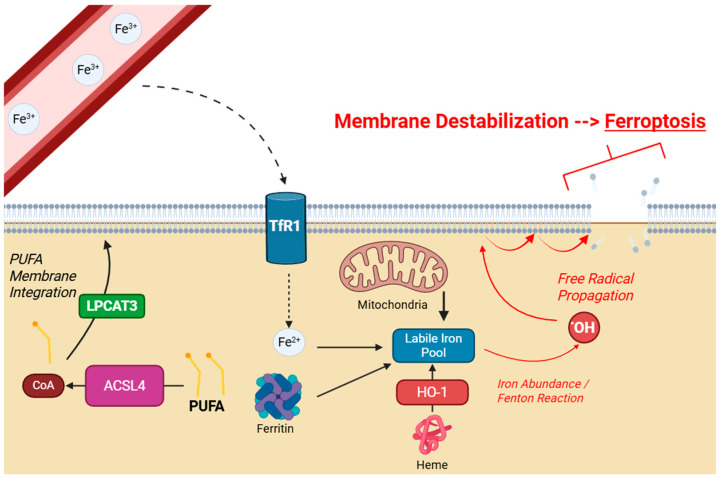
Schematic of cellular ferroptosis. Extracellular Fe^3+^ enters through TfR1 and contributes to the intracellular LIP. Other sources include ferritin, mitochondria, and heme. LIP overload drives lipid peroxidation by the Fenton reaction. PUFAs are particularly susceptible, and are integrated into the membrane by LPCAT3. Created with BioRender.com.

**Table 1 cells-14-01927-t001:** Summary of ferroptosis-inducing compounds.

Molecule	Class	Mechanism	Notable Studies
Erastin	Quinazoline derivative	SLC7A11 inhibitor	Shibata et al. (2019) [41], Li et al. (2020) [42]
Sorafenib	Kinase inhibitor	SLC7A11 inhibitor	Li et al. (2020) [42]
RSL3	GPX4 inhibitor	Direct GPX4 inhibitor	Ye et al. (2020) [9]
Tubastatin A	HDAC6 inhibitor	GPX4 destabilizer	Liu et al. (2023) [43]
ML385	NRF2 pathway inhibitor	NRF2-SCL7A11 pathway inhibitor	Yan et al. (2024) [44]
BIBR1532	Telomerase inhibitor	Enhances oxidative stress	Bao et al. (2024) [45]
Acetaminophen	Analgesic/antipyretic	Enhances oxidative stress	Gai et al. (2020) [46]
Fin56	CoQ10 synthase pathway inhibitor	GXP4 degradation, CoQ10 depletion	Sun et al. (2021) [47] Shimada et al. (2016) [48]
FINO_2_	Iron oxidizer	Iron oxidation, GXP4 inactivation	Gaschler et al. (2018) [49]
Brequinar	DHODH inhibitor	Mitochondrial lipid peroxidation	Mao et al. (2021) [50]
Artesunate	Antimalarial	Inhibition of PRDX1 and PRDX2	Liu et al. (2025) [51]
Statins	HMG-CoA reductase inhibitors	ILF3 inhibition, GSH, GPX4, and SLC7A11 downregulation	Sun et al. (2025) [52] Zhang et al. (2022) [53]

## Data Availability

Not applicable.

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
