# Peer review of "Ferroptosis and Radiotherapy in Lung Cancer"

_cells, 2025, doi:10.3390/cells14231927_

Round 1

Reviewer 1 Report

Comments and Suggestions for Authors

This review article provides a comprehensive examination of the biological mechanisms linking ferroptosis (an iron-dependent form of regulated cell death) with radiotherapy in the context of lung cancer treatment. The authors successfully synthesize current literature to explain how radiation can prime cells for ferroptosis and how this synergy might be exploited to overcome radioresistance in non-small cell lung cancer (NSCLC). A significant strength of this work is its balanced perspective; rather than focusing solely on the therapeutic benefits, the authors rightly emphasize the "double-edged sword" nature of this approach, dedicating appropriate space to the risks of radiation-induced lung injury (RILI) and systemic toxicity. The inclusion of Table 1, which summarizes ferroptosis-inducing compounds, is also a helpful resource for readers.

However, the manuscript currently suffers from a lack of polish regarding copy-editing and formatting, which distracts from the scientific content. There are several typographical errors, grammatical awkwardness, and leftover template text that need to be addressed before publication. Additionally, while the molecular mechanisms are described in detail, the clinical translation section could be strengthened by referencing specific ongoing clinical trials or discussing the impact of radiation fractionation schemes more explicitly.

Detailed comments:

  1. Section 3.1 (Radiotherapy as a Ferroptosis Inducer): The authors explain that RT induces ferroptosis, but they do not touch upon whether the dose fractionation matters. Does Stereotactic Body Radiation Therapy (SBRT/SABR) with high doses per fraction induce ferroptosis more effectively than conventional 2Gy fractions? If the literature is silent on this, the authors should add a sentence in the "Future Directions" section suggesting that optimal dosing schedules for ferroptosis induction need to be established.

  1. Section 4 (Clinical Integration): While the authors discuss the concept of clinical integration well, the review would benefit significantly from a mention of specific, active clinical trials. If there are registered trials (e.g., on ClinicalTrials.gov) combining radiotherapy with agents like sorafenib or sulfasalazine in lung cancer, citing their NCT numbers would add concrete value for clinician readers. If no such trials exist, explicitly stating this highlights a critical gap in current research.

Comments on the Quality of English Language

1. There is a significant typo in the summary (line14) where it reads "receive oxygen species." This should clearly be "reactive oxygen species." As this is in the abstract, it is the first thing a reader sees and needs to be corrected to maintain professional standards.

2. Introduction (Line 28): The sentence structure here is grammatically incorrect: "Radiotherapy (RT), and has been a cornerstone..." The word "and" is unnecessary. The authors should remove it to read, "Radiotherapy (RT) has been a cornerstone..."

3. Section 4.4 (Line 307): There is a typo at the beginning of the sentence: "Ther is potential synergy..." This should be corrected to "There is potential synergy...".

Author Response

Comments 1: Section 3.1 (Radiotherapy as a Ferroptosis Inducer): The authors explain that RT induces ferroptosis, but they do not touch upon whether the dose fractionation matters. Does Stereotactic Body Radiation Therapy (SBRT/SABR) with high doses per fraction induce ferroptosis more effectively than conventional 2Gy fractions? If the literature is silent on this, the authors should add a sentence in the "Future Directions" section suggesting that optimal dosing schedules for ferroptosis induction need to be established.

Response 1: We agree with the reviewer that a discussion of the effect of RT dosing on ferroptosis is a valuable addition to our commentary. We have added results from 2 studies which highlight this effect in non-LC cancers, and state there is a lack of investigation in LC. We also add to Section 5.1 (Limitations of Current Models), that therapeutic windows should be established by specifically comparing conventional versus high-dose RT. 

Comments 2: Section 4 (Clinical Integration): While the authors discuss the concept of clinical integration well, the review would benefit significantly from a mention of specific, active clinical trials. If there are registered trials (e.g., on ClinicalTrials.gov) combining radiotherapy with agents like sorafenib or sulfasalazine in lung cancer, citing their NCT numbers would add concrete value for clinician readers. If no such trials exist, explicitly stating this highlights a critical gap in current research.

Response 2: The idea to go over any current trials can allow readers to correlate preclinical investigations with human-based data. Unfortunately, we were only able to find 1 trial which specifically investigates a ferroptosis inducer with RT in LC, and it was terminated before publication. This highlights a strong gap, and we incorporate this point and a citation of the terminated trial in the introduction of Section 4 (Clinical Integration and Challenges).

Reviewer 2 Report

Comments and Suggestions for Authors

Prem C Patel BS and Eva Mercedes Galvan MD submitted an interesting review about ferroptosis and RT in LC. The topic was of a certain significance nowadays, and might arouse some impacts in its field. However, some issues were pending addressed. The reviewer suggested a Minor Revision for this review. Detailed comments:

  • The “structured” Abstract was a bit unnecessary. The paper was a Review, and an unstructured Abstract was okay.
  • The statistics on LC should be updated. The referenced number in the Introduction was of 2022, out of date.
  • Figure 1 and its relevant description in Section 2.1 should be reconsidered. Apart from the Fe ions from the bloodstream, the labile iron pool in the cells might be one of the most important source of Fe ions for ferroptosis.
  • More ferroptosis inducing compounds should be briefly mentioned in Section 3.2.
  • Regarding Section 5, information about nanomedicines could be supplemented.

Author Response

Comments 1: The “structured” Abstract was a bit unnecessary. The paper was a Review, and an unstructured Abstract was okay.

Response 1: We appreciate the reviewer’s perspective regarding the abstract format. We chose a structured abstract to help readers easily identify the scope and major conclusions of our review, especially given the wide breadth of basic science and clinical concepts discussed. 

Comments 2: The statistics on LC should be updated. The referenced number in the Introduction was of 2022, out of date.

Response 2: We appreciate the reviewer’s focus on including the most recent information available. Unfortunately we were unable to find any precise statistics on global cancer burden after 2022. We were able to find American numbers for incidence in 2022 and deaths in 2023 from the Center for Disease Control, which were published in June 2025 and reported as “the most recent data available”. 

Comments 3: Figure 1 and its relevant description in Section 2.1 should be reconsidered. Apart from the Fe ions from the bloodstream, the labile iron pool in the cells might be one of the most important source of Fe ions for ferroptosis.

Response 3: We agree with the reviewer’s belief that our original diagram insufficiently covered sources of intracellular iron. We have updated our image with three new sources of intracellular iron accumulation (ferritin, heme, and mitochondria). Furthermore, we have lengthened the original figure description to detail the mechanistic overview as an aid to the image. We have also included statements regarding these new iron sources in the text (Section 2.1) with relevant citations. 

Comments 4: More ferroptosis inducing compounds should be briefly mentioned in Section 3.2.

Response 4: We appreciate the reviewer’s comments. There are a diverse array of potentially useful ferroptosis inducers, and a more complete discussion increases our review’s relevance. We have included several additional compounds, and added them to our table overview as well. 

Comments 5: Regarding Section 5, information about nanomedicines could be supplemented.

  •  

Response 5: We agree that information on nanoparticles should be added to Section 5, as they represent a promising tool for clinical translation. As such, we have added information with relevant citations to Section 5.2 (Safety and Translation to Human Research). We have also decided to supplement Section 4.2 (Balancing Efficacy and Toxicity) with a deeper discussion of individual studies and nanomedicine to provide a more clear picture of where the research on these medicines currently stands.